# Constraint-Data-Value-Maximization: Utilizing Data Attribution for Effective Data Pruning in Low-Data Environments

## Abstract

Attributing model behavior to training data is an evolving research field. A common benchmark is data removal, which involves eliminating data instances with either low or high values, then assessing a model's performance trained on the modified dataset. It is generally expected that removing low-value instances results in a gradual decline in accuracy, while the removal of high-value instances leads to a sharp decrease in performance. Many existing studies leverage Shapley-based data values for this task. In this paper, we demonstrate that these data values are not optimally suited for pruning low-value data when only a limited amount of data remains. To address this limitation, we introduce the Contsraint-Data-Value-Maximization approach, which effectively utilizes data attributions for pruning in low-data scenarios. By casting pruning as a constrained optimization that both maximizes total influence and penalizes excessive per-test contributions, CDVM delivers robust performance even when only a small fraction of the data is retained. On the OpenDataVal benchmark, CDVM consistently outperforms existing alternatives, achieving state-of-the-art accuracy and competitive runtime.

## 1 Introduction

Machine learning models, especially large language models, have an insatiable demand for data, while the availability of data is stagnating. By attributing the influence of training data on model performance, the required amount of data can be reduced, thereby saving energy and improving model quality. Early works in this direction, such as influence functions (Kwon and Zou, 2021), aim to gain insights into model behavior by attributing the influence of individual training instances on test instances, thereby serving as a method for explainable AI. Conversely, methods like data Shapley (Ghorbani and Zou, 2019) have been used to assess the influence of single training instances on model performance, referring to this as data value, and applying this understanding for data removal. Typically, these approaches rely on Shapley-based methods (or approximations) to compute the value of each data instance.

Sorscher et al. (2022) benchmark methods for pruning data on ImageNet, showing that novel algorithms for data pruning can improve scaling laws, thus reducing the resource costs associated with modern deep learning.

The motivation for our work is that current data-pruning methods, especially those based on semi-values, suffer from inherent limitations that prevent them from fully exploiting the pruning potential. Semi-values are a broad class of cooperative-game-theoretic attributions, among them the Shapley value, that assign importance to each instance by averaging its marginal contributions across all subsets. We first analyze their shortcomings and then leverage our insights to design a new pruning algorithm. Our method formulates pruning as an optimization over a data attribution matrix and is evaluated on the OpenDataVal benchmark (Jiang et al., 2023), where it outperforms existing baselines, including the state-of-the-art techniques identified by Sorscher et al. (2022). Our findings show that there is still substantial room to improve data pruning, which in turn can lower training costs and reduce energy consumption. Our main contributions are:

- Using a synthetic example, we demonstrate that semi-value-based attributions allocate smaller marginal contributions to instances in larger clusters. This imbalance causes large clusters to be pruned too early, producing unbalanced removal patterns and suboptimal pruning performance.

- We demonstrate that optimal retention sets are non-nested: the subset containing the top 50% of data does not necessarily have to contain the subset of the top 30% data.

- Based on these insights, we introduce Constraint-Data-Value-Maximization (CDVM), a novel algorithm that treats data pruning as optimization problem over a data attribution matrix.

- We benchmark CDVM on six datasets from OpenDataVal, showing superior runtime and accuracy.

## 2 BACKGROUND, MOTIVATION & RELATED WORK

We begin with a concise overview of data valuation, then examine pruning and other evaluation benchmarks, outline their limitations, and finally introduce two concepts that motivate our method.

### 2.1 DATA VALUATION AND PRUNING

Data Valuation assesses the overall impact of individual training instances on the model performance, effectively answering the question, "How much did a training instance contribute to the model's performance?" The value assigned to each training instance $i$ is represented as a scalar. Consequently, the valuation scores for a dataset are expressed as a vector $v \in \mathbb{R}^n$, where $n$ is the number of training instances.

#### 2.1.1 ESTIMATING DATA VALUES

We now introduce the basic notation and the main estimation methods for data values used in this paper. For a comprehensive survey, see Hammoudeh and Lowd (2022) and Hwee et al. (2022).

- $D = \{(x_i, y_i)\}_{i=1}^n$ is a labeled dataset with inputs $x_i$ and labels $y_i$.
- $f_D$ is the model trained on the dataset $D$.
- $\theta_D$ are the corresponding model parameters.
- $f_{D \cup d_j}$ denotes a model trained on the union of $D$ and the data instance $d_j = (x_j, y_j)$.
- $\mathcal{U}$ represents a utility function, such as accuracy in a classification setting.

**Leave-One-Out**   The simplest approach to estimating the influence of a training instance is the leave-one-out (loo) method, which involves excluding a particular data instance during training and comparing the model performance or test predictions with and without this instance. This method can be approximated by influence functions (Kwon and Zou, 2021) without the need for re-training. The main limitation is that the effect of omitting a single data instance can often be obscured by the remaining data and the inherent noise in the training process (K and Søgaard, 2021). As a result, many data instance may appear to have a negligible value. Empirical evidence also suggests that loo is not effective for benchmarks in data valuation (Jiang et al., 2023). Formally, the loo-value of data instance $d_i$ can be expressed as $V(d_i) = \mathcal{U}(f_D) - \mathcal{U}(f_{D \cup d_i})$.

**Semi-value-based Estimates**   Semi-value-based techniques quantify the importance of a training instance $d_i$ by its marginal contribution over all subsets $S \subseteq D \setminus \{d_i\}$. For data valuation, three variations were proposed; original Shapley value (Ghorbani and Zou, 2019), Banzhaf (Wang and Jia, 2022), and Beta Shapley (Kwon and Zou, 2021). Technically, all these methods differ only by the weighting of each subset $w(S)$ and can be expressed as $V(d_i) = \sum_{S \subseteq D \setminus \{d_i\}} w(S) \left[ \mathcal{U}(f_S) - \mathcal{U}(f_{S \cup d_i}) \right].$

These methods generally outperform the loo estimate in practice. However, their main drawback is their exponential computational complexity. To mitigate this, Monte Carlo or other sampling-based techniques are often used to approximate data values. Notably, data Banzhaf (Wang and Jia, 2022) has proved to be computationally efficient due to the *Maximum Sample Reuse (MSR)* principle. The data value is approximated by sampling subsets $S \subset D$ of the training data with probability $p$ and training a model on each subset. This process is repeated multiple times, and the data value of data instance $d_i$ is computed as the performance difference between subsets where $d_i$ is included versus where it is not.

**Out-of-Bag and Memorization Estimates** The concept of memorization has been introduced in recent studies, wherein a training instance $i$ is considered "rare" if its exclusion from the training set significantly reduces the probability that $i$ is correctly classified by the same model (Feldman, 2020b; Paul and Dziugaite). A related method used in data valuation is the out-of-bag estimate, known as *DataOob*, where the significance of training instances is assessed using out-of-bag samples (Kwon and Zou, 2023). In each iteration, the training set is split into in-bag and out-of-bag groups, a model is trained on the in-bag samples, and predictions are made on the out-of-bag samples. The value of a data instance is then determined based on its memorization score during these out-of-bag assessments. Although these techniques are not suited for data attribution (as no test set is involved), they have proven effective in data pruning tasks, even on ImageNet (Sorscher et al., 2022).

### 2.1.2 Data Pruning and Benchmarks for Data Valuation

Data-valuation methods are commonly evaluated on three tasks:

1. **Noise Detection:** Identify and remove corrupted or mislabeled examples, which tend to carry large negative value due to their disruptive effect on training (Jiang et al., 2023).

2. **Domain Transfer:** Select a subset of source–domain data that maximizes accuracy on a target–domain test set (e.g., choosing MNIST digits to improve performance on street-number datasets) (Ghorbani and Zou, 2019).

3. **Data Removal:** Measure how model accuracy changes when portions of the training set are removed in order of increasing or decreasing value. Removing high-value instances first should cause a steep accuracy drop, whereas pruning low-value instances should have minimal impact.

In this work, we focus on the third task, data removal, specifically pruning low-value data, since it directly addresses the practical goal of reducing dataset size without sacrificing performance. From here on, we use *data pruning* to mean the removal of low-value points. In the literature, authors differ in whether they report results for removing low-value data (pruning) or for removing high-value data first. For instance, the original Data Shapley study (Ghorbani and Zou, 2019) presents low-value pruning curves, while the OpenDataVal framework (Jiang et al., 2023) emphasizes high-value removal. We are not aware of any formal discussion explaining this discrepancy. Empirically, memorization-based or out-of-bag-based methods tend to excel at low-value pruning, whereas Shapley-based techniques often show stronger effects when high-value data is removed first.

### 2.1.3 Limitations of Data Values for Data Pruning

After briefly reviewing the main approaches to data valuation, we now highlight their shortcomings in the context of data pruning. To support the illustration, consider the dataset in Figure 1 (a). It consists of two Gaussian clusters per class with centers $\mu_1 = (-2, 0.5), \mu_2 = (2.5, 0)$ (red) and $\mu_3 = (-2.5, -0.5), \mu_4 = (2, 0)$ (blue). In total there are eight instances: three in $\mu_1$, two in $\mu_2$, two in $\mu_4$, and one in $\mu_3$. The test set comprises only the four cluster centers. The black line shows the decision boundary learned by an multi-layer perceptron. Importantly, removing any entire cluster shifts this boundary dramatically (Figure 1 (b)). Appendix C displays the same setup in more detail.

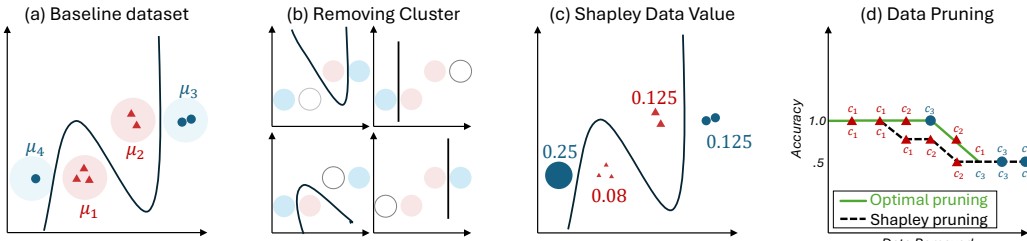

Figure 1: (a) Baseline synthetic dataset comprising 8 points from 4 clusters. (b) Illustrates the changed decision boundary after removing an entire cluster. In each scenario, the decision boundary undergoes significant alterations. (c) Displays the Shapley data value. (d) Test accuracy as we iteratively remove instances (x-axis: removal step 1–8; y-axis: accuracy). In the optimal (green) removal order, clusters are pruned as $c_1, c_1, c_2, c_3, c_2, c_1, c_3, c_4$, whereas the Shapley-based (black) order is $c_1, c_1, c_1, c_2, c_2, c_{1,3}, c_4$ and $c_i$ belongs to the $i$-th cluster.

**1. LOO has Redundancy Bias and Attributes Non-zero Value Only to Unique Data**   We begin with the observation that loo attributions reward only non-redundant samples. In Figure 1 (a), only the singleton instance in $\mu_4$ receives a nonzero value of 0.25 (Figure 1 (c)), since its removal introduces a change in the decision boundary (Figure 1 (b)), causing a test error at the respective cluster center. All other instances receive a value of zero due to their redundancy and can therefore be pruned in any order.

**2. Semi-Value-Based Techniques Scale with Cluster Size and Cause Imbalanced Pruning**   Semi-values (e.g., Shapley, Banzhaf) allocate each instance's importance inversely to its redundancy: the more neighbors an instance has, the smaller its marginal contribution (see Figure 1(c)). Consequently, large clusters are completely pruned first, which initially removes redundant examples but then triggers a steep accuracy drop as soon as any cluster is depleted (Figure 1(d)).

This effect can be also observed if we move to real data. In the left plot of Figure 2, we compare DataBanzhaf against random pruning on CIFAR-10, using either 1 000 or 10 000 models to estimate data values. Both data Banzhaf variants outperform random removal up to about 50 % pruning. Beyond that instance, the 10 000-model variant plunges significantly below the random baseline, whereas the 1 000-model variant continues to slightly outperform random pruning, even though the larger ensemble should, in principle, yield more accurate attributions.

**3. Pruning Subsets Are Not Nested**   Finally, we observe that optimal retention sets at different pruning levels are not nested: the subset that maximizes accuracy for one budget $s$ may exclude instances that are essential for another budget $s' \neq s$. In Figure 2 (center), we use our own method to identify the best subsets for retaining 5%, 10% and 15% of the data. We then perform sequential pruning of the remaining instances, always keeping the preselected subset intact and plot test accuracy versus fraction removed. Each accuracy curve peaks exactly at its target retention level (dots), and even a slight deviation from that budget causes a dramatic collapse in performance. A similar pattern appears for Memorization/DataOob (Figure 2, right): removing the highest-value instances first (red curve) initially improves performance before it plummets, whereas retaining those same instances until the very end yields almost state-of-the-art final accuracy. This mirrors the finding of Sorscher et al. (2022), namely that the examples most dispensable in data-rich regimes are precisely those that must be kept when data become scarce. For further intuition on these phenomena, see the synthetic example in Appendix C.5.

These observations highlight the need for a pruning strategy that (i) tracks influence at the level of individual test samples and (ii) can flexibly re-optimize for each pruning budget. To that end, we now review two key building blocks of our approach: data attribution and influence-function–guided pruning.

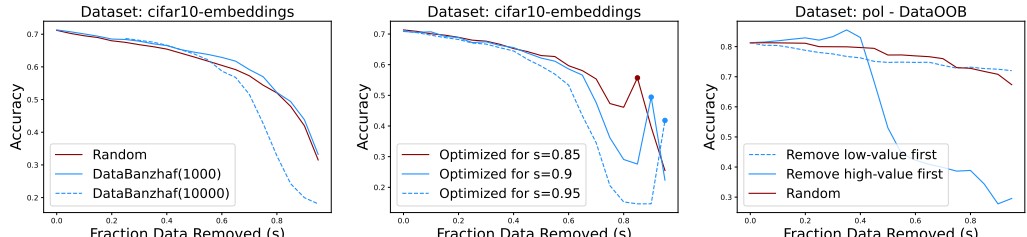

Figure 2: All plots show test accuracy as a function of the fraction of training data removed. **Left:** CIFAR-10 results for DataBanzhaf pruning. We estimate Banzhaf values with 1.000 models (solid blue) and 10.000 models (dashed blue), and compare against random removal (solid red). Both Banzhaf variants outperform random up to 50% pruning, but counterintuitively the 10.000-model variant degrades faster than the 1.000-model version. **Center:** An extreme example of non-nested pruning subsets. Each curve is optimized for exactly 85%, 90%, or 95% removal (i.e., 15%, 10%, 5% retention). Accuracy peaks precisely at the target rate (dots), and removing more or less data causes a steep collapse. **Right:** Memorization/DataOob pruning. The dashed blue curve removes lowest-value instances first; the solid blue curve removes highest-value instances first; and the red curve is random removal. Surprisingly, the very instances whose early removal boosts accuracy (and outperforms random) when data is abundant, must be retained until the end under high removal budgets to again outperform the random baseline.

## 2.2 Relationship of Core-Sets, Active Learning and Valuation/Pruning

Core-set selection methods (Feldman, 2020a) aim to choose a small subset that preserves the geometry or distribution of the full dataset, typically by minimizing a covering or clustering objective.

Active learning (Settles, 2010) sequentially selects unlabeled examples to label by maximizing model uncertainty or expected error reduction. In prior work, active learning has also been benchmarked as a pruning baseline (Sorscher et al., 2022).

In contrast, data valuation and pruning methods assign importance scores to training instances via game-theoretic attributions. Hence, although core-set and active learning methods also select representative subsets, their methodology differs from attribution-based pruning, which targets test influence.

## 2.3 Preliminaries: Data Attribution & Influence-Function Pruning

We continue by introducing two fundamental concepts, data attribution and influence-function pruning, before presenting our method.

### 2.3.1 Data Attribution

Data attribution is conceptually related to data valuation, but traces the influence of individual train instances down to specific test samples. The influence of training data on test predictions is quantified using the attribution matrix $\mathbf{T} \in \mathbb{R}^{n \times m}$, where $n$ is the number of train instances and $m$ the number of test instances. A high value of $\mathbf{T}_{i,j}$ indicates that the train instance $i$ significantly impacts the prediction for test instance $j$. The connection between both is that data values can be estimated by averaging over the rows (test instances) of $\mathbf{T}$, formulated as $v_i = \frac{1}{m} \sum_{k=0}^{m} \mathbf{T}_{i,k}$. This per–test–sample breakdown provides a fine-grained view of dataset contributions, which we leverage directly in our method. Several methodologies have been developed to estimate this influence, with influence functions being one of the pioneering approaches (Koh and Liang, 2017). More recently, TRAK has emerged as a scalable method for data attribution across large datasets (Park et al., 2023).

### 2.3.2 INFLUENCE-FUNCTION–GUIDED PRUNING

Yang et al. (2022) cast data pruning as a discrete optimization problem over binary selection variables, with the goal to minimize overall parameter change. Let $w \in \{0,1\}^n$ be the indicator vector specifying which of the $n$ training samples are retained. The parameter change from removing a single instance $d_i$ is given by the influence function $\mathcal{I}(d_i) = \theta_{D \setminus d_i} - \theta_D \approx \frac{1}{n} H_\theta^{-1} \nabla_\theta \mathcal{L}(d_i; \theta_D)$, where $H_\theta$ is the Hessian of the total training loss at $\theta_D$. For a subset of instances, these influences simply add up. Define the matrix $\mathbf{Z} = \begin{bmatrix} \mathcal{I}(d_1), \ldots, \mathcal{I}(d_n) \end{bmatrix}$, so that the total parameter change of the selected subset is $\mathbb{S}\,w$. They then solve

$$\min_{w \in \{0,1\}^n} \left\| \mathbf{Z}\,w \right\|_2 \quad \text{s.t.} \quad \sum_{i=1}^n w_i = S \,,$$

where $S$ is the desired subset size. Although this method achieves strong empirical performance and inspired our approach, it has two major drawbacks. First, it requires (approximate) Hessian inversion for every training instance, which is computationally expensive. Second, because it relies on influence functions, essentially approximating leave-one-out, it inherits loos's limitations (see Sec. 2.1.3): removing a single, redundant instance is expected to produce a negligible influence score (K and Søgaard, 2021).

## 3 SIZE-CONSTRAINED DATA-VALUE-MAXIMIZATION: OPTIMIZING DATA VALUES FOR PRUNING

Building on the inspiration from Yang et al. (2022) and the limitations identified in Section 2.1.3, we introduce a novel method to derive data values optimized for pruning. In Section 2.1.3, we observed that semi-value-based data values fail at pruning because they tend to remove entire clusters first. To overcome this, we leverage the attribution matrix

$$\mathbf{T} \in \mathbb{R}^{n \times m},$$

which describes the influence of each of the $n$ training samples on each of the $m$ test samples. A naive way to derive pruning scores from $\mathbf{T}$ is to average over its columns, but this approach suffers (among other issues) from the cluster-removal limitation noted above. Instead, $\mathbf{T}$ provides fine-grained, per-test influence values that do not suffer from redundancy bias. We leverage this to ensure balanced coverage: at each pruning step, no test sample (and thus no implicit cluster) should have zero total influence. To formalize this, let

$$w \in \{0,1\}^n$$

be the binary indicator vector selecting exactly $S$ out of the $n$ training instances. The induced utility vector for the $m$ test samples is

$$v = \mathbf{T}^\top w \ \in \ \mathbb{R}^m \,, \quad v_j = \sum_{i=1}^n \mathbf{T}_{ij}\,w_i \ .$$

A naive pruning objective would be

$$\max_w \sum_{j=1}^m v_j \quad \text{s.t.} \quad \sum_{i=1}^n w_i = S, \quad w_i \in \{0,1\}.$$

This objective maximizes total influence but can still concentrate all value on a few test instances. To ensure balanced coverage, we introduce nonnegative slack variables $t_j$ that caps any excess above a threshold $\kappa$. In other words, any amount $\max\{v_j - \kappa, 0\}$ is transferred into $t_j$ and subtracted from the objective. We call the resulting formulation Constrained Data-Value Maximization (CDVM):

$$\max_{w,t} \quad \alpha \sum_{j=1}^{m} v_j \; - \; (1-\alpha) \sum_{j=1}^{m} t_j \; ,$$

$$\text{s.t.} \quad v = \mathbf{T}^\top w \; ,$$

$$\sum_{i=1}^{n} w_i = S \; ,$$

$$t_j \geq 0 \; , \quad j = 1, \dots, m \; ,$$

$$t_j \geq v_j - \kappa \; , \quad j = 1, \dots, m \; ,$$

$$w_i \in [0,1] \; , \quad i = 1, \dots, n \; .$$

This formulation directly remedies the shortcomings identified in Section 2.1.3 by (1) maximizing each test sample's total influence via $\mathbf{T}$, thereby avoiding redundancy bias; (2) penalizing any excess above $\kappa$, thus ensuring every test cluster retains influence; and (3) enforcing a fixed subset size $S$ to identify the optimal subset for the given budget. Furthermore, because all constraints are linear and some decision variables are integer-valued, the problem can be formulated as a mixed-integer linear program.

## 3.1 Implementation Details

In our final setup, we relax the binary constraint $w_i \in \{0,1\}$ to a continuous one $w_i \in [0,1]$. This converts the mixed-integer program into a pure linear program, greatly improving tractability and without any observable loss in our experiments. The algorithm takes as input the attribution matrix $\mathbf{T}$ and introduces two hyperparameters:

- $\alpha$: non-negative trade-off between total utility and penalty for exceeding $\kappa$,
- $\kappa$: soft upper bound on the influence per test sample.

Computing $\mathbf{T}$ is the main computational bottleneck, since it requires retraining models on sampled subsets, an expense shared by all semi-value-based methods. Consequently, any parameter used to estimate $\mathbf{T}$ effectively becomes a hyperparameter. Here, we follow the Maximum Sample Reuse (MSR) principle of Ye et al. (2023):

1. Sample $T$ subsets $S_t \subseteq D$ by including each training instance with probability $p$.
2. Train a model on each $S_t$ and record the performance (or indicator of correct classification) on each test instance.
3. Estimate $\mathbf{T}_{ij}$ as the average difference in that performance for test instance $j$ when $d_i$ is in versus out of $S_t$.

In our experiments, we set $p = 0.03$ and $T = 10{,}000$, ensuring each training instance appears often enough for stable estimates. The entries $\mathbf{T}_{ij} \in [-1, 1]$ are easily interpretable: $-1$ means "always causes a mistake" and $+1$ means "always ensures correct prediction." Moreover, $\mathbf{T}$ is sparse, most training instances have zero or negligible influence on most test instances, which significantly accelerates subsequent optimization.

Once $\mathbf{T}$ is computed, we solve the relaxed CDVM problem using the Disciplined Parametrized Programming framework. This formulation enables caching, so we can quickly resolve the program after it has been solved once. This efficiency allows a lightweight grid search over the two hyperparameters $\alpha$ and $\kappa$.We then run the optimization independently for each retained fraction (e.g., 30%, 25%).

## 4 Experimental Results

We evaluate CDVM on the six datasets from the OpenDataVal benchmark (Jiang et al., 2023). By default, each dataset is subsampled to 1,000 training, 500 validation, and 500 test examples to match prior work (e.g. Wang and Jia, 2022; Jiang et al., 2023) and reduce

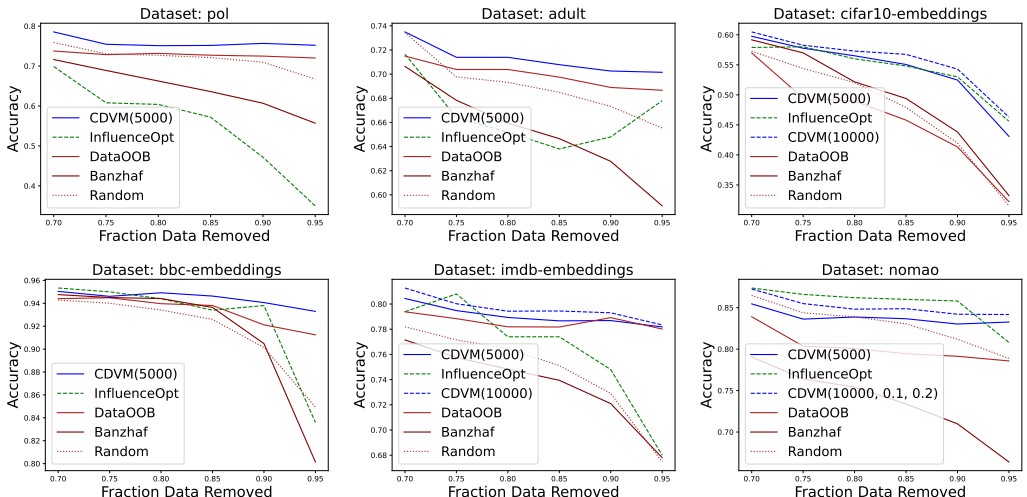

Figure 3: Accuracy on 30%, 25%, 20%, 15%, 10%, and 5% of remaining training data for six datasets in the OpenDataVal benchmark Jiang et al. (2023). We utilized a sampling probability of $p = 0.03$ for computing the attribution matrix and automatically optimized parameters for the CDVM method. Out of 36 configurations, CDVM achieved state-of-the-art performance in 28 setups.

computational cost. To demonstrate that CDVM extends beyond this regime, we include in Appendix B.3 a scaling experiment on the Fashion-MNIST dataset (60,000 train, 10,000 test). Thanks to the sparsity of the attribution matrix $\mathbf{T}$, the CDVM optimization solves in 10–30 minutes, including hyperparameter grid-search, while the retraining cost to estimate $\mathbf{T}$ (shared by all semi-value methods) takes several hours, showing that CDVM also scales to larger data sizes. We compute the attribution matrix $\mathbf{T}$ on the training–validation split and use it to select (prune) training instances. Final performance is then assessed on the held-out test set. Each experiment is run with 25 random seeds, and we report the average test accuracy when retaining 5%, 10%, 15%, 20%, 25%, and 30% of the training data. We compare against the following baselines:

- **Random** removal of training samples.
- **DataOob/memorization** identified as state-of-the-art method (Kwon and Zou (2023); Sorscher et al. (2022)).
- **DataBanzhaf** (Wang and Jia, 2022), a semi-value–based method grounded in the MSR principle, which we also employ.
- **Influence Optimization** (Yang et al. (2022)). We encountered some stability issues with the original code: e.g. optimizing for a 10% final subset occasionally performed better when using the budget for 5%. To be fair, at each pruning level we compare against the best accuracy achieved by this method over any budget. We also relaxed the constraint $w_i \in \{0, 1\}$ to a continuous one $w_i \in [0, 1]$, since the original mixed-integer-programming formulation often failed to converge. Consequently, these results should be viewed as an upper bound on the method's performance.

We restrict our benchmark to these methods because they have proven effective in prior work. DataBanzhaf serves as a baseline to ensure any performance gains stem from our optimization rather than the attribution algorithm. For CDVM, we fix the sampling probability at $p = 0.03$ and train $T = 5000$ models (the primary computational bottleneck). In some cases, tuning $p$ or increasing $T$ yields gains; we also report those as dashed line when they are significant.

Figure 3 summarizes results over the 36 evaluation instances (6 datasets $\times$ 6 pruning rates), our default CDVM configuration (solid blue) outperforms all baselines in 24 cases. Per-dataset tuning (dashed blue) yields some gains and increases the total number of state-of-the-art results to 28. Appendix A provides the full tabular breakdown including standard deviation.

Among all baselines, only the method of Yang et al. (2022) (green dashed line) outperforms CDVM, and this occurs mainly on the `nomao` dataset, where it edges out CDVM at 5 of the 6 pruning levels. It also performs competitively on `cifar10` but falls short on `pol` and `adult`, despite being evaluated as an upper bound. On the other two text datasets (`bbc` and `imdb`), its gains are occasional and much smaller. In contrast, CDVM is the only method that consistently beats the random baseline across all six benchmarks. DataOob/memorization remains competitive on `imdb` and `bbc` datasets, but never achieves state-of-the-art accuracy.

The `nomao` dataset exhibits unusual dynamics for CDVM and Yang et al. (2022)'s methods. With default hyperparameters, CDVM initially underperforms random pruning up to an 85% removal rate. We found that manually tuning to $p = 0.1$, $\kappa = 0.2$, $\alpha = 0.1$ restores its advantage. Likewise, Yang et al. (2022)'s approach attains its best scores on `nomao` only when its ranked instances are removed first and the remainder are kept at random, i.e., by applying its ranking in reverse. We attribute CDVM's initial failure mode on this dataset to the high proportion of near-zero entries in the attribution matrix $\mathbf{T}$. Increasing the sampling probability $p$ seems to improve this, and using a smaller threshold $\kappa$ prevents CDVM from assigning too much influence to individual test instances when overall attributions are small.

### 4.1 ABLATION STUDY

**Runtime Comparison**   Figure 4(a) plots each method's average runtime per experiment against its normalized performance (scaled to [0,1] across all 36 evaluation settings). A score of 1 denotes the top accuracy in every setting, while 0 denotes the worst. Details on the metric and detailed training and optimization times are provided in Appendix B. CDVM achieves the best speed–accuracy trade-off, outperforming the baselines by a wide margin. Interestingly, in this aggregate view DataOob/memorization outperforms Influence Optimization in overall efficiency, despite our earlier finding that Influence Optimization beats DataOob on individual datasets, this is because DataOob delivers consistently strong (though not state-of-the-art) accuracy with much lower computational cost.

**Hyperparameter Sensitivity**   Although CDVM introduces two hyperparameters ($\alpha$ and $\kappa$), we find them robust across tasks and can be set without a grid search. In practice we recommend

$$\alpha = 0.5, \quad \kappa = \max_{i,j} \mathbf{T}_{ij} \; + \; |S| \operatorname{mean}_{i,j}(\mathbf{T}_{ij}),$$

which adapts $\kappa$ automatically to the dataset and retention budget $S$. In Figure 4(a) we compare the original CDVM(5k) with grid-searched hyperparameters against CDVM(5k, default) using the settings above. The performance difference is marginal ($\leq 0.05$ in normalized performance) while the default configuration incurs zero search overhead, making it a practical choice for most applications.

**Rank Correlation**   To quantify non-nestedness, we compute the average Spearman rank correlation between the instance-importance rankings at different retention levels, across all seeds and datasets (Figure 4(d)). Correlation declines as the gap between budgets widens, confirming that optimal subsets diverge for different removal rates. Interestingly, the diagonal entries (same budget, different seeds) show higher correlation for smaller subsets, suggesting that tight budgets admit fewer combinations, whereas larger subsets offer more redundancy and hence greater ranking variability.

## 5   SCALABILITY AND COMPUTATIONAL COST

In Appendix B.3 we present a scaling experiment on the full Fashion–MNIST dataset (60 000 train, 10 000 test). Despite this larger size, CDVM remains efficient: thanks to the sparsity of the attribution matrix $\mathbf{T}$, retaining only 5–10% of its entries yields a solve time of 10–30 minutes, whereas retraining models to estimate $\mathbf{T}$ (shared by all semi-value approaches) requires several hours.

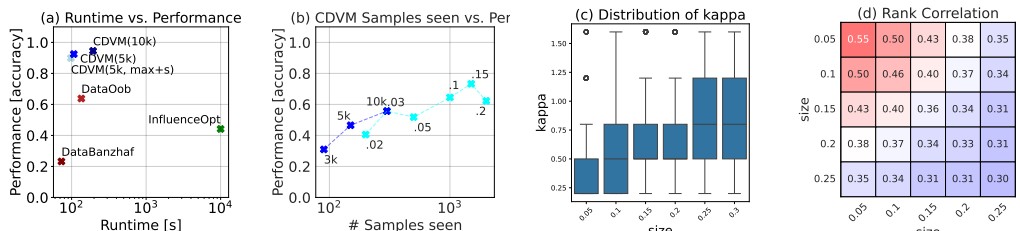

Figure 4: **(a)** Runtime vs. normalized performance for all benchmarked methods, aggregated over six datasets and six pruning levels. **(b)** CDVM performance as a function of how often each sample is seen during training for sampling probability $p$ (cyan) and number of models trained $T$ (blue). **(c)** Distribution of the selected slack threshold $\kappa$ across datasets and retention fractions. **(d)** Spearman rank correlation between instance-importance rankings at different pruning budgets, showing decreasing correlation for more distant subset sizes.

Extending CDVM to even larger datasets such as ImageNet introduces two main bottlenecks: (i) estimating $\mathbf{T}$ in terms of compute and memory, and (ii) solving the linear program at that scale. For context, prior studies have precomputed influence or memorization estimates on ImageNet by training thousands of ResNet-50 models[1], but never shared the full attribution matrix ($\approx 250\,\text{GB}$ for train×test). By thresholding $\mathbf{T}$ to keep only its top 10% nonzeros, this can be reduced to $\approx 25\,\text{GB}$; further memory savings are possible via half-precision floats.

On the computational side, retraining to estimate $\mathbf{T}$ can be alleviated by recent estimators such as TRAK (Park et al., 2023), which avoid retraining thousands of models at the cost of a different approximation. If the resulting $\mathbf{T}$ is still too large for a single linear program solve, one can partition it into smaller blocks and solve the optimization iteratively on these chunks. We leave these extensions to future work.

## 6 Summary, Limitations & Outlook

In this work, we introduced Constraint-Data-Value-Maximization (CDVM), an optimization-based framework that leverages the data-attribution matrix $\mathbf{T}$ to prune low-value examples in low-data regimes. We demonstrated competitive accuracy and runtime across six Open-DataVal tasks. However, since the entries of $\mathbf{T}$ are not additive, CDVM may miss higher-order interactions. Integrating Shapley interaction indices (Muschalik et al., 2024) could capture these effects, albeit with additional computational overhead. Finally, CDVM relies on a selected soft upper bound $\kappa$ and incurs quadratic cost in computing and storing $\mathbf{T}$ (e.g., roughly 250 GB for a naive implementation without sparsity on the full ImageNet-1k train and val splits), which might limit scalability. Future work could mitigate these bottlenecks by employing attribution estimators such as TRAK Park et al. (2023), exploiting sparsity or low-rank structure in $\mathbf{T}$, or solving the optimization on partitioned submatrices, offering opportunities for future extensions with only modest computational overhead increases.

### Reproducibility statement

To ensure full reproducibility, we will release all source code on GitHub upon publication. During the review period, we provide a standalone Jupyter notebook that computes the data-attribution matrix $\mathbf{T}$, formulates and solves the CDVM optimization, and prints results against a random baseline. The notebook is self-contained and can be applied to any dataset.

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

## Use of Large Language Models (LLMs)

LLMs were primarily used to enhance the paper's language and support code completion during implementation, as well as to define, refine, and improve the optimization problem. Although the initial concept originated with the authors, LLMs contributed significant refinements and performance optimizations.

## A  RESULT DETAILS

We provide supplementary details for the main paper. Table 1 tabulates the numerical results underlying the benchmark curves, while Figures 5 and 6 plot CDVM's performance and runtime across different values of $p$ and $T$.

| | 30% Data | 25% Data | 20% Data | 15% Data | 10% Data | 5% Data |
|---|---|---|---|---|---|---|
| | | | *nomao* | | | |
| CDVM5k | $0.855 \pm 0.02$ | $0.836 \pm 0.02$ | $0.839 \pm 0.02$ | $0.837 \pm 0.02$ | $0.830 \pm 0.02$ | $0.833 \pm 0.03$ |
| CDVM10k | $0.848 \pm 0.01$ | $0.840 \pm 0.02$ | $0.833 \pm 0.02$ | $0.839 \pm 0.02$ | $0.830 \pm 0.02$ | $0.821 \pm 0.02$ |
| CDVM-n | $0.872 \pm 0.02$ | $0.855 \pm 0.02$ | $0.848 \pm 0.02$ | $0.849 \pm 0.02$ | $0.842 \pm 0.02$ | $\mathbf{0.842} \pm 0.02$ |
| InfOpt | $\mathbf{0.874} \pm 0.00$ | $\mathbf{0.866} \pm 0.00$ | $\mathbf{0.862} \pm 0.00$ | $\mathbf{0.860} \pm 0.00$ | $\mathbf{0.858} \pm 0.00$ | $0.808 \pm 0.00$ |
| DataOOB | $0.839 \pm 0.01$ | $0.804 \pm 0.01$ | $0.801 \pm 0.01$ | $0.794 \pm 0.01$ | $0.791 \pm 0.01$ | $0.786 \pm 0.02$ |
| Banzhaf | $0.791 \pm 0.02$ | $0.764 \pm 0.02$ | $0.754 \pm 0.02$ | $0.734 \pm 0.03$ | $0.710 \pm 0.03$ | $0.664 \pm 0.07$ |
| Random | $0.865 \pm 0.02$ | $0.844 \pm 0.02$ | $0.839 \pm 0.02$ | $0.830 \pm 0.03$ | $0.812 \pm 0.03$ | $0.789 \pm 0.04$ |
| | | | *cifar10* | | | |
| CDVM5k | $0.598 \pm 0.02$ | $0.578 \pm 0.02$ | $0.565 \pm 0.03$ | $0.551 \pm 0.02$ | $0.525 \pm 0.03$ | $0.431 \pm 0.03$ |
| CDVM10k | $\mathbf{0.605} \pm 0.02$ | $\mathbf{0.583} \pm 0.03$ | $\mathbf{0.573} \pm 0.03$ | $\mathbf{0.567} \pm 0.02$ | $\mathbf{0.543} \pm 0.02$ | $\mathbf{0.463} \pm 0.03$ |
| InfOpt | $0.579 \pm 0.02$ | $0.580 \pm 0.00$ | $0.560 \pm 0.00$ | $0.548 \pm 0.00$ | $0.530 \pm 0.00$ | $0.456 \pm 0.00$ |
| DataOOB | $0.570 \pm 0.01$ | $0.495 \pm 0.01$ | $0.490 \pm 0.02$ | $0.458 \pm 0.03$ | $0.413 \pm 0.10$ | $0.322 \pm 0.13$ |
| Banzhaf | $0.592 \pm 0.02$ | $0.570 \pm 0.03$ | $0.522 \pm 0.08$ | $0.494 \pm 0.08$ | $0.438 \pm 0.08$ | $0.332 \pm 0.06$ |
| Random | $0.573 \pm 0.02$ | $0.544 \pm 0.03$ | $0.520 \pm 0.03$ | $0.479 \pm 0.06$ | $0.420 \pm 0.07$ | $0.315 \pm 0.07$ |
| | | | *pol* | | | |
| CDVM5k | $0.786 \pm 0.02$ | $0.755 \pm 0.03$ | $\mathbf{0.751} \pm 0.03$ | $0.752 \pm 0.03$ | $\mathbf{0.757} \pm 0.03$ | $0.752 \pm 0.04$ |
| CDVM10k | $\mathbf{0.796} \pm 0.02$ | $\mathbf{0.760} \pm 0.03$ | $0.751 \pm 0.03$ | $\mathbf{0.752} \pm 0.03$ | $0.749 \pm 0.03$ | $\mathbf{0.753} \pm 0.03$ |
| InfOpt | $0.699 \pm 0.01$ | $0.608 \pm 0.00$ | $0.604 \pm 0.00$ | $0.572 \pm 0.00$ | $0.472 \pm 0.00$ | $0.350 \pm 0.00$ |
| DataOOB | $0.738 \pm 0.03$ | $0.729 \pm 0.04$ | $0.732 \pm 0.03$ | $0.727 \pm 0.03$ | $0.725 \pm 0.04$ | $0.720 \pm 0.04$ |
| Banzhaf | $0.716 \pm 0.04$ | $0.689 \pm 0.04$ | $0.663 \pm 0.05$ | $0.636 \pm 0.05$ | $0.607 \pm 0.04$ | $0.557 \pm 0.06$ |
| Random | $0.759 \pm 0.03$ | $0.731 \pm 0.03$ | $0.727 \pm 0.03$ | $0.721 \pm 0.04$ | $0.710 \pm 0.04$ | $0.668 \pm 0.05$ |
| | | | *imdb* | | | |
| CDVM5k | $0.804 \pm 0.01$ | $0.795 \pm 0.02$ | $0.789 \pm 0.02$ | $0.787 \pm 0.02$ | $0.787 \pm 0.02$ | $0.782 \pm 0.02$ |
| CDVM10k | $\mathbf{0.813} \pm 0.01$ | $0.800 \pm 0.02$ | $\mathbf{0.794} \pm 0.03$ | $\mathbf{0.794} \pm 0.01$ | $\mathbf{0.793} \pm 0.01$ | $\mathbf{0.783} \pm 0.02$ |
| InfOpt | $0.794 \pm 0.01$ | $\mathbf{0.808} \pm 0.00$ | $0.774 \pm 0.00$ | $0.774 \pm 0.00$ | $0.748 \pm 0.00$ | $0.680 \pm 0.00$ |
| DataOOB | $0.794 \pm 0.01$ | $0.788 \pm 0.02$ | $0.782 \pm 0.02$ | $0.782 \pm 0.02$ | $0.789 \pm 0.01$ | $0.780 \pm 0.01$ |
| Banzhaf | $0.772 \pm 0.03$ | $0.758 \pm 0.03$ | $0.748 \pm 0.03$ | $0.739 \pm 0.02$ | $0.721 \pm 0.04$ | $0.678 \pm 0.04$ |
| Random | $0.782 \pm 0.02$ | $0.772 \pm 0.02$ | $0.765 \pm 0.02$ | $0.751 \pm 0.03$ | $0.729 \pm 0.03$ | $0.675 \pm 0.05$ |
| | | | *adult* | | | |
| CDVM5k | $\mathbf{0.735} \pm 0.01$ | $\mathbf{0.714} \pm 0.01$ | $\mathbf{0.714} \pm 0.02$ | $\mathbf{0.708} \pm 0.01$ | $0.703 \pm 0.02$ | $\mathbf{0.702} \pm 0.01$ |
| CDVM10k | $0.726 \pm 0.02$ | $0.709 \pm 0.01$ | $0.707 \pm 0.01$ | $0.706 \pm 0.01$ | $\mathbf{0.705} \pm 0.02$ | $0.694 \pm 0.01$ |
| InfOpt | $0.717 \pm 0.01$ | $0.664 \pm 0.00$ | $0.652 \pm 0.00$ | $0.638 \pm 0.00$ | $0.648 \pm 0.00$ | $0.678 \pm 0.00$ |
| DataOOB | $0.715 \pm 0.01$ | $0.704 \pm 0.01$ | $0.704 \pm 0.01$ | $0.698 \pm 0.01$ | $0.689 \pm 0.01$ | $0.687 \pm 0.01$ |
| Banzhaf | $0.706 \pm 0.02$ | $0.678 \pm 0.02$ | $0.659 \pm 0.02$ | $0.647 \pm 0.03$ | $0.628 \pm 0.03$ | $0.591 \pm 0.04$ |
| Random | $0.734 \pm 0.02$ | $0.698 \pm 0.02$ | $0.693 \pm 0.02$ | $0.685 \pm 0.02$ | $0.673 \pm 0.02$ | $0.655 \pm 0.03$ |
| | | | *bbc* | | | |
| CDVM5k | $0.950 \pm 0.01$ | $0.946 \pm 0.01$ | $\mathbf{0.949} \pm 0.00$ | $0.946 \pm 0.01$ | $0.941 \pm 0.01$ | $0.933 \pm 0.01$ |
| CDVM10k | $0.946 \pm 0.01$ | $0.947 \pm 0.01$ | $0.947 \pm 0.01$ | $\mathbf{0.947} \pm 0.01$ | $\mathbf{0.943} \pm 0.01$ | $\mathbf{0.934} \pm 0.01$ |
| InfOpt | $\mathbf{0.953} \pm 0.01$ | $\mathbf{0.950} \pm 0.00$ | $0.944 \pm 0.00$ | $0.934 \pm 0.00$ | $0.938 \pm 0.00$ | $0.836 \pm 0.00$ |
| DataOOB | $0.944 \pm 0.00$ | $0.945 \pm 0.00$ | $0.940 \pm 0.00$ | $0.938 \pm 0.00$ | $0.921 \pm 0.01$ | $0.912 \pm 0.01$ |
| Banzhaf | $0.948 \pm 0.01$ | $0.945 \pm 0.01$ | $0.944 \pm 0.01$ | $0.936 \pm 0.01$ | $0.905 \pm 0.05$ | $0.801 \pm 0.16$ |
| Random | $0.943 \pm 0.01$ | $0.940 \pm 0.01$ | $0.934 \pm 0.01$ | $0.926 \pm 0.02$ | $0.901 \pm 0.05$ | $0.849 \pm 0.06$ |

Table 1: Accuracy on 30%, 25%, 20%, 15%, 10%, and 5% of training data for six datasets in the OpenDataVal benchmark Jiang et al. (2023). Out of 36 configurations, CDVM achieved state-of-the-art performance in 28 setups. The Error margins represent standard deviations based on 25 experiments.

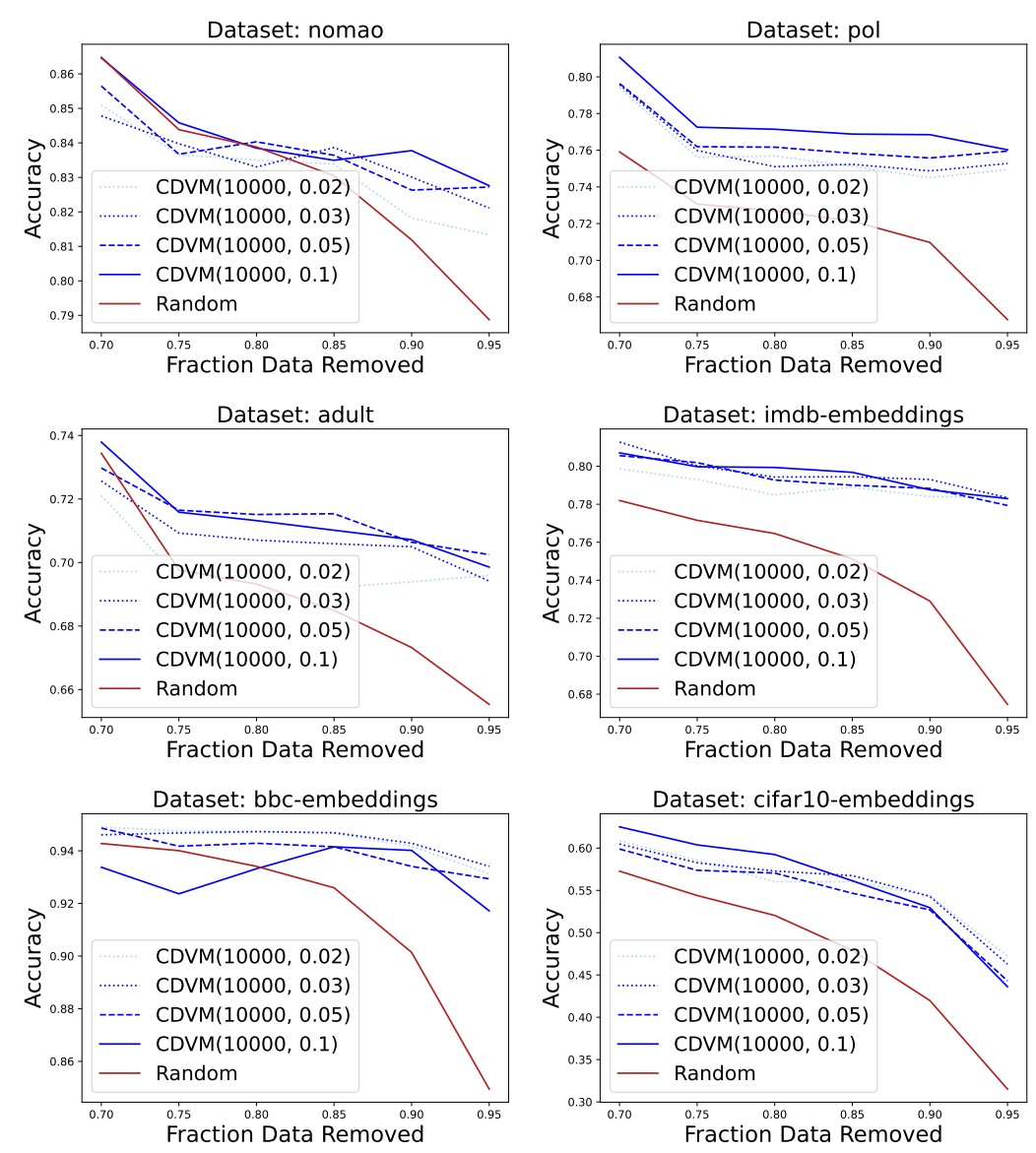

Figure 5: CDVM performance for different sampling probabilities $p \in \{0.02, 0.03, 0.05, 0.10\}$ on five datasets. A higher sampling rate ($p = 0.10$) yields the best pruning accuracy on Nomao, POL, and Adult, and outperforms lower $p$ values up to 85% removal on CIFAR-10, but degrades performance on BBC.

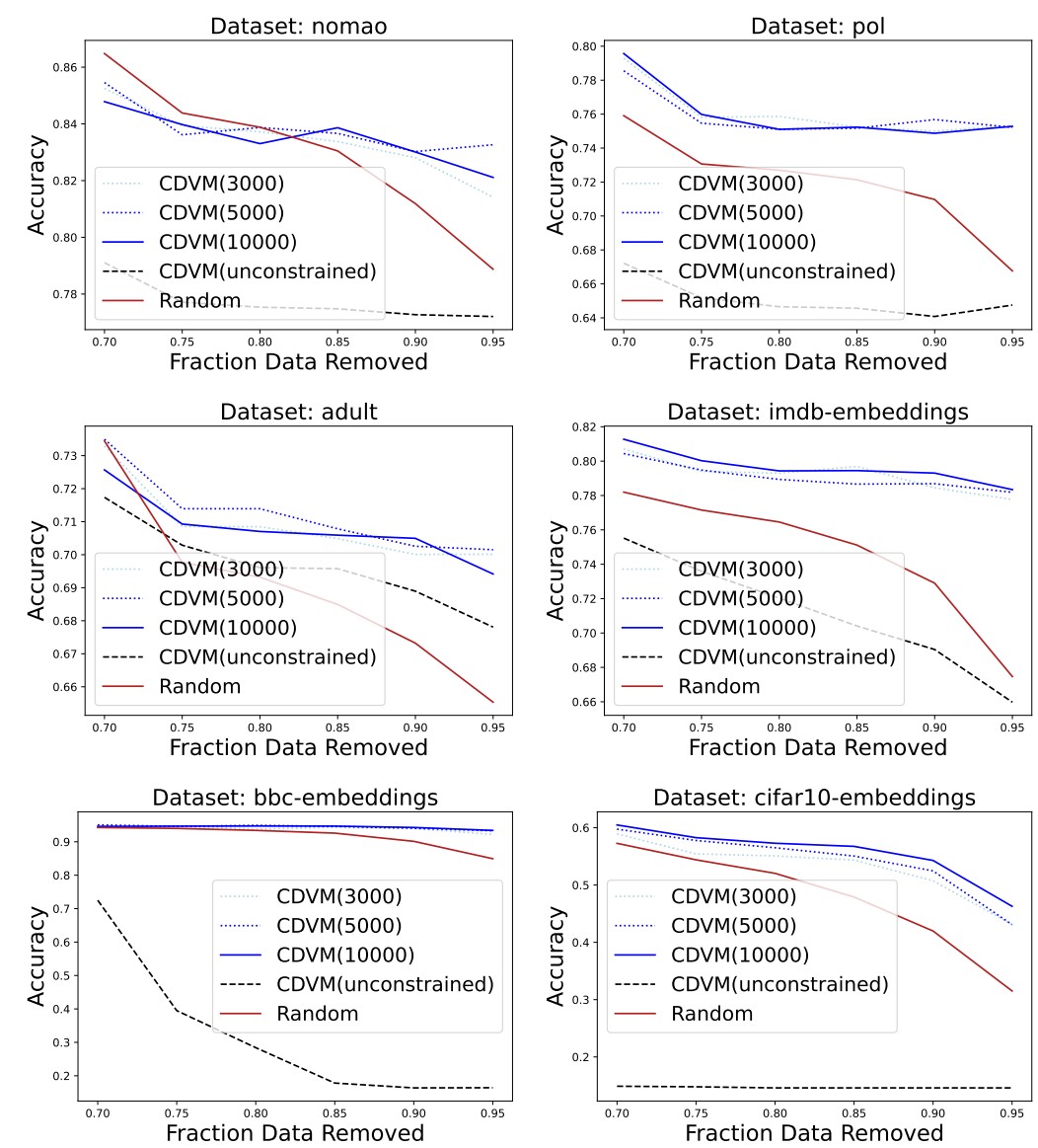

Figure 6: Effect of model-count and slack constraint on CDVM's runtime and accuracy. For each dataset, we compare CDVM using 3 000, 5 000, and 10 000 models to estimate the attribution matrix $\tau$, as well as a variant without the $\kappa$ constraint. In general, increasing the number of models improves pruning quality at the cost of longer runtime, while removing the slack constraint causes a severe drop in performance.

# B  Ablation Study Details

In addition, we detail our ablation study, particularly the performance normalization procedure, provide deeper insight into algorithm runtimes by distinguishing preparation and optimization times, and demonstrate CDVM's scalability on a full dataset.

## B.1  Performance Normalization

In Figure 4, we condense each method's performance across all evaluation settings into a single normalized score. To do so, we normalize each method's total score by the sum of the best- and worst-case performances. Formally, let $S$ be the set of all 36 evaluation settings (6 datasets $\times$ 6 pruning levels), and let $p_{m,s}$ denote the test accuracy of method $m$ on setting $s$. Define

$$P_m = \sum_{s \in S} p_{m,s}, \qquad P_{\max} = \sum_{s \in S} \max_{m'} p_{m',s}, \qquad P_{\min} = \sum_{s \in S} \min_{m'} p_{m',s}.$$

Then the normalized performance of method $m$ is

$$\widetilde{P}_m = \frac{P_m - P_{\min}}{P_{\max} - P_{\min}},$$

which maps the aggregate score of each method into the interval $[0, 1]$.

## B.2 Optimization and Runtime

| Method | Preparation Time | Optimization Time | Total Time |
|--------|------------------|-------------------|------------|
| DataOob | 135 s (1000 models) | n.a. | 135 s |
| CDVM | 97 s (5000 models) | 10 s | 107 s |
|  | 184 s (10000 models) |  | 194 s |
| InfOpt | 6 h (1000 data instances) | 366 s (cifar-10) | ≈ 6h |
|  |  | 3 s (imdb) | ≈ 6h |

Table 2: Runtime comparison (preparation + optimization). Wall-clock times for (i) DataOob/memorization, (ii) CDVM, and (iii) Influence-Function Optimization (InfOpt) on the OpenDataVal benchmark. DataOob uses bootstrap samples of size $n$ (with replacement) and retrains $T_{\text{OOB}} = 1,000$ models. CDVM samples each training point with probability $p = 0.03$ and retrains $T_{\text{CDVM}} = 5,000$ (or 10,000) models to achieve stable estimates. Because each CDVM model sees only 3% of the data, individual training runs are much faster. InfOpt avoids retraining but must invert a Hessian per training instance and solve a quadratic program, resulting in multi-hour runtimes. Our method has constant optimization time because all datasets are scaled to the same size (1000 training and 500 validation + test instances). For InfOpt, optimization time scales with the dataset's input dimension, whereas preparation time remains largely constant.

## B.3   Scaling

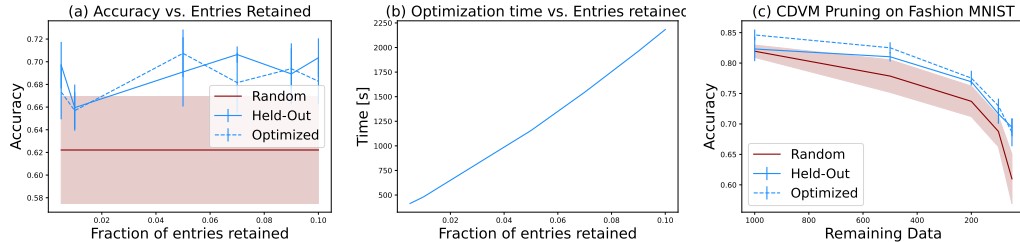

Figure 7: **(a)** Test accuracy for different sparsity cut-offs in the attribution matrix, expressed as the percentage of entries retained by not setting them to zero. **(b)** Runtime (in seconds) for solving the CDVM optimization on each corresponding sparse matrix. **(c)** CDVM test performance compared against a random-pruning baseline.

So far, we have evaluated CDVM on the OpenDataVal benchmark by subsampling each split to $1\,000/500/500$ (train/validation/test), enabling rapid comparisons across methods. To assess scalability on a larger attribution matrix, we apply CDVM to Fashion-MNIST ($60\,000$ training and $10\,000$ test images). Fashion-MNIST was chosen because it offers a sizable dataset while still permitting fast model training. As before, the resulting attribution matrix $\tau$ is highly sparse: many training instances have zero or negligible influence on most test samples. However, due to numerical noise, most entries remain small nonzero values and must be filtered out.

To study the effect of this residual noise, we threshold $\tau$ by retaining only the top $0.5\,\%$–$10\,\%$ of its entries and setting all others to zero. The corresponding test accuracies are shown in Figure 7. We split the test set into an evaluation partition, for selecting training examples and tuning hyperparameters, and a held-out test partition for final performance assessment, and report results on both.

As shown, retaining just $5\,\%$ of the examples suffices to achieve high accuracy; including more examples yields no further benefit. Estimating $\tau$ requires retraining between $5\,000$ and $8\,000$ models, which takes approximately $1$–$10$ hours on a standard workstation, depending on the subsampling rate $p$.

## C   SYNTHETIC DATASET

In this section, we provide further details and empirical examples on the synthetic dataset and Shapley-value data valuation, illustrating how interactions among data points can induce non-monotonic pruning behavior, for example, we construct a dataset in which removing more examples paradoxically improves accuracy, so that keeping less data can outperform keeping more.

**Full dataset**

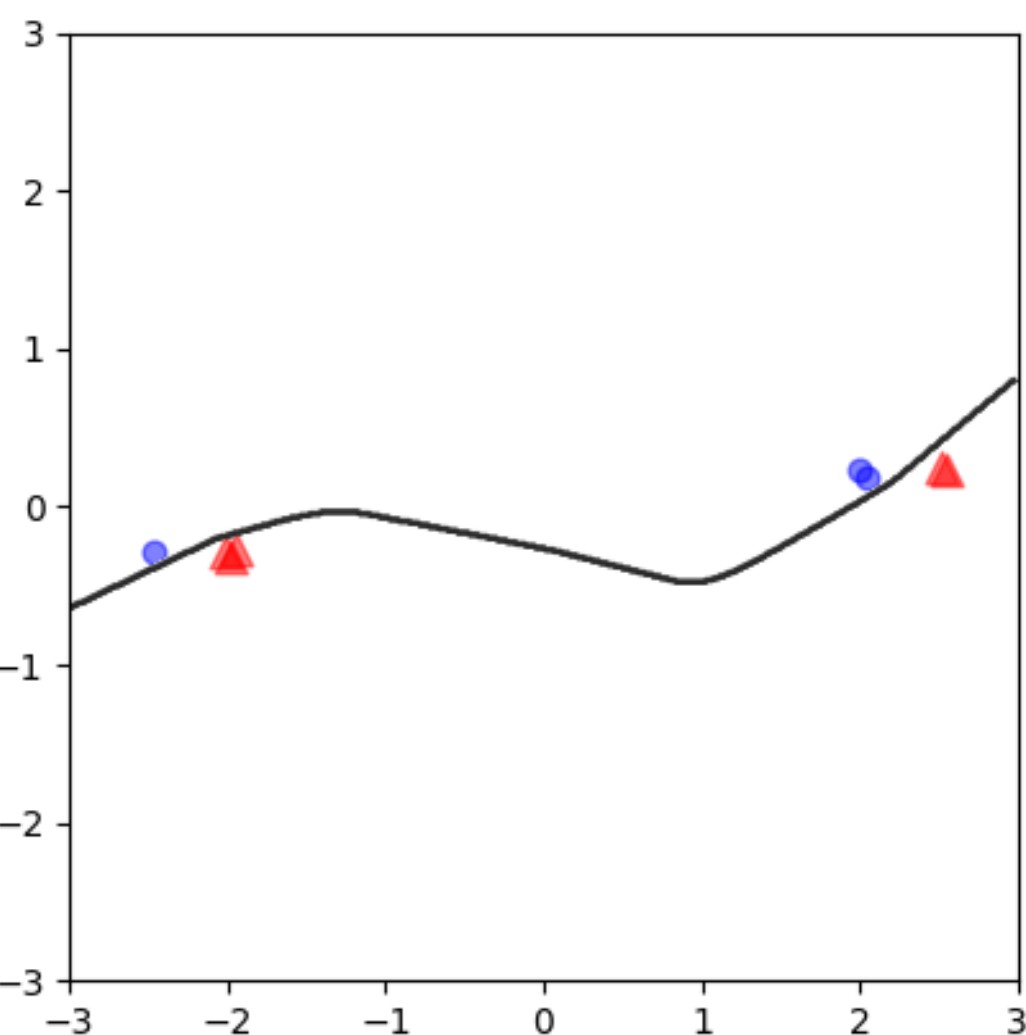

Figure 8: Synthetic dataset with eight points from four clusters. All clusters except the blue one centered at $(-2.5, -0.5)$ contain more than one point.

## C.1 Removing Entire Clusters

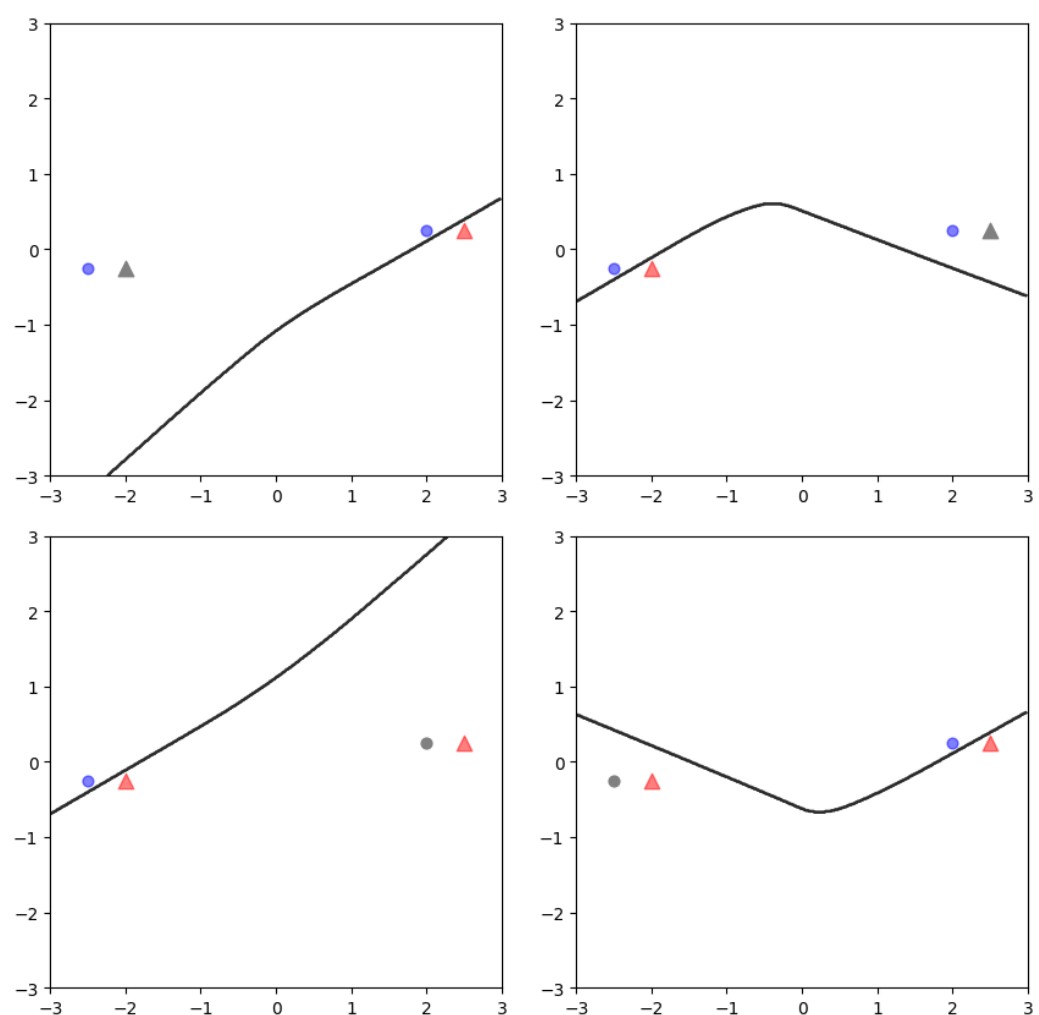

Figure 9: Effect of removing an entire cluster. The removed cluster is grey.

## C.2   LEAVE ONE OUT

**LOO left red cluster**

**LOO right red cluster**

**LOO right blue cluster**

**LOO left blue cluster**

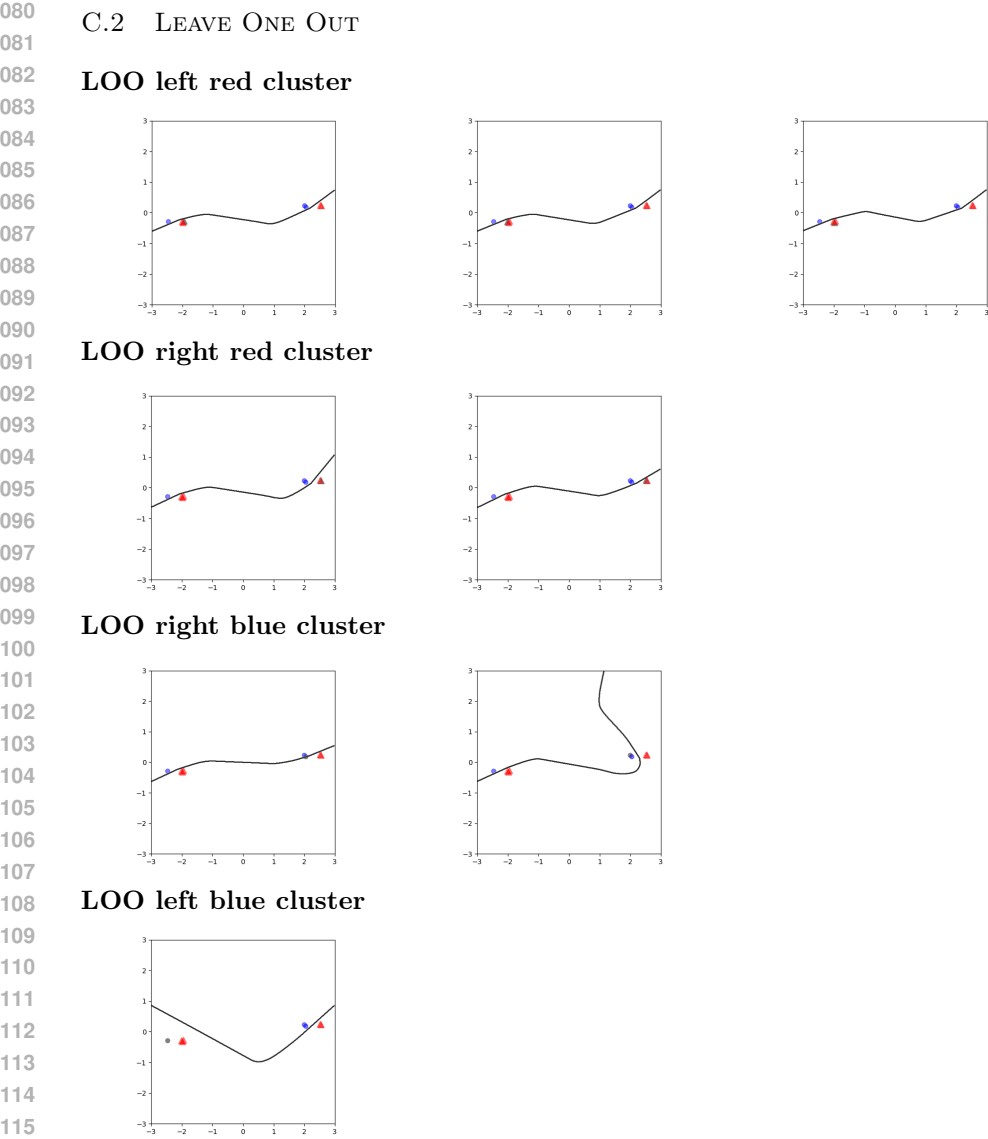

Figure 10: Leave-one-out (LOO) on the dataset from Figure 8. All clusters except the last contain more than one point; therefore, the decision boundary remains unchanged when a point is removed from these clusters. Each plot shows the effect of removing exactly one point from the respective cluster. Consequently, only the point from the left blue cluster will exhibit a non-zero leave-one-out data value.

## C.3 SHAPLEY DATA VALUE

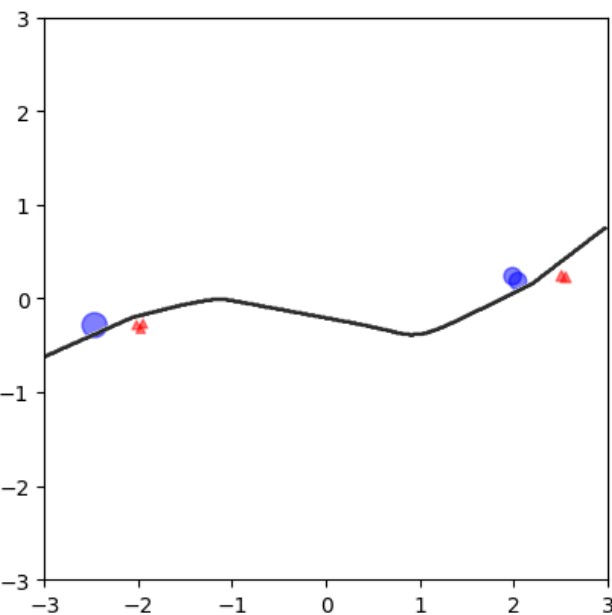

Figure 11: Shapley data valuation scores for the synthetic dataset. The black line represents the decision boundary of an MLP trained on this dataset. Given the 256 possible combinations for subsets, not all are plotted. Instead, the plot displays the computed data Shapley values, where the size of a point indicates its value. As observed, the values are proportional to the cluster size, with the blue singleton point exhibiting the highest value.

## C.4 SHAPLEY-BASED PRUNING VS. OPTIMAL PRUNING

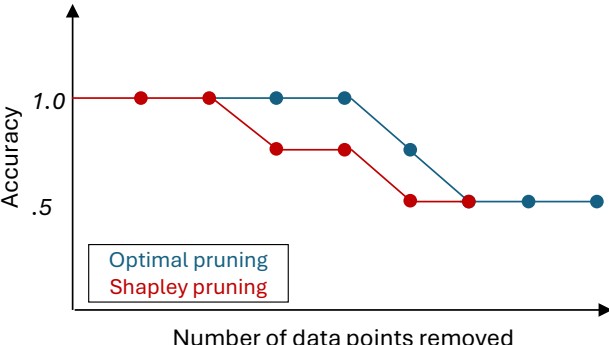

Figure 12: Difference between data pruning based on Shapley values (red) and the optimal pruning (red) on the synthetic dataset. Each dot in the plot represents one data point being removed. Shapley-based pruning will initially remove the three red points with lowest value. Once the last one of them is removed, the accuracy drops. In contrast, in the optimal pruning we can remove two points from the cluster with three points and one from the two clusters with two points without loosing any performance.

## C.5 Data Interactions

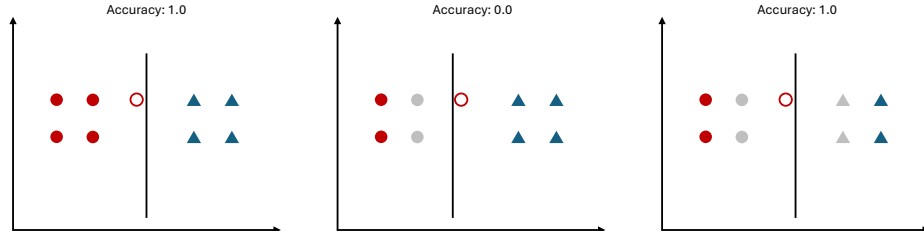

Figure 13: Synthetic example of interaction effects when removing data. In this case, the black line represent the decision boundary of a support vector machine. We assume that the data value is simply the distance to the decision boundary and the red circle the only point in the test. It is labeled "red". The left-most plot shows the original dataset. The red circle is on the right side of the decision boundary and, hence, classified correctly by the support vector machine. In the center plot, two red points were removed. As a consequence, the decision boundary shifts and the red circle is miss-classified. In the right-most plot, two blue points were additionally removed. The decision boundary is back at its original position and the circle is correctly classified again. This shows how data values (and pruning) depend on data interactions and that model performance during pruning is not necessarily monotonic.

