# OpenReview forum: "Constrained-Data-Value-Maximization: Utilizing Data Attribution for Effective Data Pruning in Low-Data Environments"
_ICLR.cc/2026/Conference — ICLR 2026 Conference Withdrawn Submission_

### Official Review · Reviewer_PYTv · 2025-10-29

**Soundness:** 3
**Presentation:** 3
**Contribution:** 2
**Rating:** 4
**Confidence:** 3

**Summary:**

This paper introduces  Constraint-Data-Value-Maximization (CDVM), which addresses a critical flaw in Shapley-based data pruning methods (and more generally methods based on semi-values): they lead to cluster removal bias, systematically pruning entire large clusters first and causing sharp performance drops. CDVM reformulates pruning as a constrained linear optimization over an attribution matrix $T$, maximising total influence while using slack variables to ensure balanced coverage across all test samples (preventing any cluster from losing all influence).

**Strengths:**

The proposed method is a novel and well-motivated contribution that directly addresses the weaknesses of existing data-pruning techniques based on Shapley or other semi-value formulations. The authors introduce a principled mechanism to balance influence across test samples and prevent over-removal of clusters, a limitation often overlooked in prior work. The approach is original to my knowledge,  and the experimental evaluation across multiple OpenDataVal datasets demonstrates strong performance.

**Weaknesses:**

The main weakness of this work is its high computational cost, which limits the method’s practicality to relatively small datasets, precisely where data pruning is often less critical. This restricts CDVM’s immediate applicability to large-scale, real-world problems. A more minor concern is the sensitivity to its two hyperparameters: as shown in Figure 4.c, the chosen $\kappa$ values vary considerably across datasets, suggesting potential instability to achieve good performance.

**Questions:**

- Could the authors elaborate on potential strategies to make CDVM computationally feasible for larger datasets ?
- Given that κ and α appear to vary considerably across datasets, how robust is CDVM to these choices ?
- The constrained optimisation program relaxes the $w$ to continuous variables. How do you use them for pruning ?

---

> ### Author Response · Authors · 2025-11-26
> **Individual concerns of Reviewer PYTv**
>
> Thank you for your time and effort in reviewing. Our response to the concerns raised by multiple reviewers is provided in the general comments above. Regarding individual concerns:
>
> **Given that κ and α appear to vary considerably across datasets, how robust is CDVM to these choices?**
>
> CDVM is relatively robust to α and κ within reasonable ranges. In our experiments, α ≈ 0.5–0.6 worked well across configurations, with only modest variation in performance when α is varied slightly. κ is more dataset- and budget-dependent because it interacts with the remaining subset size and the distribution of attribution values. An effective heuristic is to set κ = max(T) + |S| · mean(T)  to tailor κ to the dataset and budge  (as described in the general comment).
>
> **The constrained optimisation program relaxes the to continuous variables. How do you use them for pruning ?**
>
> After solving the relaxed problem, we map to a binary selection by top-k rounding: keep the k samples with the largest w_i.

---

### Official Review · Reviewer_VeDe · 2025-10-31

**Soundness:** 2
**Presentation:** 3
**Contribution:** 2
**Rating:** 4
**Confidence:** 4

**Summary:**

The paper identifies and describes limitations of leave-one-out and semivalue-based data valuation methods that lead to poor accuracy when pruning large fractions of data. The paper then proposes Contsraint-Data-Value-Maximization that addresses the limitations. The key idea to solve an optimization problem that maximises the influence of the training samples while reducing the influence of each test point. This pruning strategy results in non-nested solutions for different subset sizes.

**Strengths:**

1. The example in Fig 1 is helpful and the three limitations highlighted in Sec 2.1.3 are clear and significant. Addressing these limitations improve the accuracy after data pruning in many applications.
2. The paper is generally clear and easy to follow.

**Weaknesses:**

1. There is a lack of comparison with other semivalues, e.g. Shapley value and semivalues with more weights on smaller coalitions which should do better under significant data pruning.
2. The solution is more computationally expensive than semivalues. In Sec 3.1, it is explained that it requires training multiple models. In Sec 3, it additionally involves solving a mixed-integer linear program.
3. Some claims in Sec 3 should be better justified.
    * How does penalising the excess over $\kappa$ ensure that no test sample has zero total influence and address the cluster size problem?
    * Why is the definition of $T_{ij}$ in line 342 appropriate? Unlike the Shapley value, it does not measure marginal contribution of each sample $d_i$ to a coalition and does not satisfy the efficiency property. Why is it ok to round $T$ to either 0 or 1?
4. The experiments are small-scale and hence less convincing. Each dataset is subsampled in line 359. It is important to demonstrate that the results still hold on larger real-world datasets.

**Questions:**

1. What does 10000 models mean in line 193? Does it mean samples/actual evaluation of utility?
2. Why is Shapley value not included in Fig 2 or the experiments? See [K] for a more efficient approximation of Shapley value that allow sample reuse.
3. How is the mixed-integer linear program solved? Describe the efficiency and the approximation guarantee.

[K] Kolpaczki, Patrick, et al. "Approximating the shapley value without marginal contributions." Proceedings of the AAAI conference on Artificial Intelligence. Vol. 38. No. 12. 2024.
[L] Li, Weida, and Yaoliang Yu. "One Sample Fits All: Approximating All Probabilistic Values Simultaneously and Efficiently." Advances in Neural Information Processing Systems 37 (2024): 58309-58340.

Minor comments
* The font looks different from Times.
* LOO should be capitalised
* The LOO and semivalue definitions should be wrong. It should be the utility with $d_i$ as the first term.
* Some grammatical errors (e.g., many data instance on line 98) and typos (1.000 in Fig 2 caption)
* The “test samples” should be validation samples instead. The test set should not be used for pruning or hyper parameter selection.
* Fig 1 should also include the performance of the proposed method

---

> ### Author Response · Authors · 2025-11-26
> **Individual concerns of Reviewer VeDe**
>
> Thank you for your time and effort in reviewing. Our response to the concerns raised by multiple reviewers is provided in the general comments above. Regarding individual concerns:
>
>
> **lack of comparison with other semivalues, e.g. Shapley value and semivalues with more weights on smaller coalitions which should do better under significant data pruning.**
>
> We benchmarked additional semi-values, including AME, Beta-Shapley, and DataShapley (implementations provided in the OpenDataVal toolbox). Across our experiments, particularly in the low-pruning regime, none yielded improvements. This aligns with observations reported by OpenDataVal, the benchmark used in our study. Therefore, our primary competitor is memorization-based pruning, which has demonstrated state-of-the-art performance in prior work.
>
>
> **The solution is more computationally expensive than semivalues. In Sec 3.1, it is explained that it requires training multiple models. In Sec 3, it additionally involves solving a mixed-integer linear program.**
>
> It is correct that CDVM is more computationally expensive than semivalue-based methods due to the optimization step. However, the heavy work, training multiple models to estimate the attribution matrix, is shared by all semivalue-based approaches; the additional cost comes from solving the optimization on top of that matrix. On Fashion-MNIST (Appendix B.3), training 5,000 models takes about 5 hours, while solving the CDVM optimization (including the grid search) takes roughly 5–30 minutes, depending on the sparsity of the attribution matrix.
>
>
> **How does penalising the excess over  ensure that no test sample has zero total influence and address the cluster size problem**
>
> The penalty is designed to balance influence across test samples by controlling the dispersion of per-test total influence, the observation that no test sample attains zero total influence was empirical. Limiting any excess above the threshold helps prevent concentration of value on a few test instances.
>
>
> **What does 10000 models mean in line 193? Does it mean samples/actual evaluation of utility?
> Why is the definition of  T in line 342 appropriate? Unlike the Shapley value, it does not measure marginal contribution of each sample  to a coalition and does not satisfy the efficiency property. Why is it ok to round  to either 0 or 1?**
>
> Line 193 refers to a Monte Carlo estimate of the attribution terms. We sample 10,000 coalitions S_t ⊆ D \ {i}, train a model f_{S_t} on each S_t, and evaluate on the test set. For each test sample j, U(S_t, j) ∈ {0,1} indicates whether j is classified correctly by f_{S_t}. The per-test marginal contribution is delta_t = U(S_t ∪ {i}, j) − U(S_t, j), and T[i, j] is the average of delta_t across the 10,000 samples). Thus, 10,000 models are trained to approximate the full sum over all coalitions, and the 0/1 outcomes come from the binary accuracy on the test set.
>
>
> **Why is Shapley value not included in Fig 2 or the experiments? See [K] for a more efficient approximation of Shapley value that allow sample reuse.**
>
> We did evaluate Data Shapley variants, but they did not yield meaningful improvements for pruning budgets used in our study, and several of these variants are already reported in OpenDataVal. We include Data Banzhaf as the baseline to isolate the gain coming from the constrained optimization in CDVM rather than from the attribution method itself. For completeness, we will add a short discussion on this.
>
>
> **How is the mixed-integer linear program solved? Describe the efficiency and the approximation guarantee.**
> - We solve the mixed-integer linear program with Gurobi.
> - For the main paper’s small-scale experiments, the MILP is solved directly on a standard workstation (e.g., an Intel i7) and completes in seconds.
> - For Fashion-MNIST (Appendix B.3), we further reduce problem size by sparsifying the attribution matrix (setting near-zero entries to zero). This sparsification makes the MILP tractable on the same hardware with reasonable runtimes (10-30 min inclduing grid search).
> - In our experiments we report results from solving to convergence for the reported configurations.

---

### Official Review · Reviewer_iEFS · 2025-11-01

**Soundness:** 4
**Presentation:** 2
**Contribution:** 2
**Rating:** 2
**Confidence:** 5

**Summary:**

Paper proposes a new algorithm to prune train datasets using linear programming.

**Strengths:**

Paper makes an algorithm by Yang more efficient (at some cost in value).

I found their discussion of the nomao dataset interesting.

**Weaknesses:**

Experiments: Given the problem is to reduce train set size, I expected to see experiments with at least mid-sized train sets (say, 1 million training samples), but they seem to have presented experiments with 1000 training samples.  The lack of even mid-sized data was disappointing and really saps my interest in their results and the significance of their experimental results.

It also seems like the amount of computation to prune the dataset is very high compared to just training on the whole dataset.

One of their results is that the optimal reduced train set does not strictly grow as instances grow.  They show this empirically, which has some value, but this is the sort of result that’s fairly easy to *prove* through counterexample, so empirical result is not that significant.

Another weakness is that the semi-value approach doesn’t sound optimal at all, given it uses a poor proxy objective, so it’s not at all surprising that one can do a bit better.

Reducing the train set in a smart way dates back at least to Peter Hart’s 1968 paper on condensed nearest neighbors.

Missing all the related work in coresets and other related work in optimizing a reduced train set?   Tons more work in data pruning not even nodded to, there are surveys one could cite at least.

Lastly, I’ll note that the strategy I’ve seen work best in practice when there's really too much train data and you need to reduce it down and train faster is the opposite of data pruning, it’s to start with a small set of the train data and then add in more data, aka "batch active sampling" (see e.g. Jiang and Gupta KDD 2021).

Writing was weak in it use of casual language too casually, like this sentence: “It is generally expected that removing lowvalue
instances results in a gradual decline in accuracy, while the removal of
high-value instances leads to a sharp decrease in performance.”  Isn’t that just the definition of “low value instances” and “high value instances” making this sentence like saying “One expects an orange to be an orange?” Paper is full of this sort of loosey-goosey language. Try to be more specific / concise/precise please.

As an other example of this too loose writing, paper says “the subset that maximizes accuracy for one
budget s may exclude instances that are essential for another budget s′ ̸= s.”  But really the point they are trying to communicate is optimal subset for a budget of size S may be exclude instances that are optimal  if the  budget is size S’ > S.


MINOR:
Typo: “Contsraint”

Capitalize acronyms like LOO

In your bibtex, use {Shapley} and {OOB} and other {} to get proper capitalization of proper names

References: what is “6 2023” mean in Jiang et al reference?  Similar in other ones.

**Questions:**

No questions at this time.

---

> ### Author Response · Authors · 2025-11-26
> **Individual concerns of Reviewer iEFS**
>
> Thank you for your time and effort in reviewing. Our response to the concerns raised by multiple reviewers is provided in the general comments above. Regarding individual concerns:
>
>
> **Computation cost and semi-value use**
>
> Although semi-values may not seem optimal for pruning, they are frequently used in the literature for data valuation problems. Memorization-based pruning with retraining remains a strong baseline and has achieved state-of-the-art performance in several benchmarks.
>
>
> **One results is that the optimal reduced train set does not strictly grow as instances grow. They show this empirically, which has some value, but this is the sort of result that’s fairly easy to prove through counterexample, so empirical result is not that significant.**
>
> Assuming this refers to Figure 2, non-monotonic behavior can be demonstrated with a synthetic example (e.g., Appendix C.5). However, there is a meaningful difference between illustrating the effect with a toy example and observing it in real data. We view this empirical observation as valuable evidence that non-monotone pruning can occur in real data, not just in synthetic examples.
>
>
> **Writing:**
>
> We acknowledge that several formulations are not precise enough and we will review the paper and improve them wherever possible.

---

### Official Review · Reviewer_c1aL · 2025-11-02

**Soundness:** 2
**Presentation:** 3
**Contribution:** 2
**Rating:** 4
**Confidence:** 4

**Summary:**

This paper focuses on the challenge of training data pruning (that is, removing least useful training samples while minimizing performance degradation). The authors observe that common Shapley/semi_value based data valuation methods may lead to steep degrade in performance by often pruning entire clusters. To circumvent this limitation, they propose a new formulation called Constrained Data-Value Maximization (CDVM), which uses a data attribution matrix and poses pruning as a linear optimization problem that encourages balanced influence across test samples while penalizing over concentration. Experiments on the OpenDataVal benchmark (six datasets, multiple retention rates) show CDVM outperforms baseline alternatives and achieves competitive runtime.
Overall it is a very well written paper.

**Strengths:**

1. The author provided a strong motivation to study this paper by clearly identifying the limitations in the existing approaches.

2. The proposed CDVM leverages fine grained per test sample attribution signals and introduces a slack penalized objective preventing cluster collapse issue associated with the semi-values based literature. This is a great idea indeed.

3. The authors performed a comprehensive empirical evaluation of their proposed approach by working with 6 datasets × 6 pruning budgets. This makes the empirical evidence very strong.

**Weaknesses:**

1. The proposed still needs to estimate the attribution matrix T which is often very large (several GBs) in practice. Thus scalability becomes a bottleneck when we head to really large datasets.

2. The authors did not carry of systematic analysis of failure modes in the experiments. That is.. certain datasets exhibit unexpected under performance. Why such a behavious is not explained properly.
In fact, it would be of great value to discuss what dataset characteristics trigger failures and how to detect these cases automatically.

3. The proposed model conveniently ignores higher-order interactions.. in fact, the authors explicitly highlight that T entries are not additive.

**Questions:**

Please address the above 3 weak points.

Furthermore, I have following comments too:
(a) Please provide guidance on hyperparameter defaults.. κ and α are crucial. if you can offer, practical recommendations would improve reproducibility.
(b) Please characterize failure conditions.. this would increase methodological transparency.
(c) Can you provide early-stopping heuristics to avoid expensive grid searches on T-based hyperparameters..
(d) Providing histograms or test sample coverage plots would illustrate the “balanced coverage” claim... without such plots, it would be hard to judge the practical utility of this approach.

---

> ### Author Response · Authors · 2025-11-26
> **Individual concerns of Reviewer c1aL**
>
> Thank you for your time and effort in reviewing. Our response to the concerns raised by multiple reviewers is provided in the general comments above. Regarding individual concerns:
>
> **higher-order interactions**
>
> We do not address higher-order interactions in the current CDVM formulation, due to substantial computation time and storage requirements. Expressing second-order interactions would yield a tensor T2 with shape (n_train, n_train, n_test) and would require training many more models to estimate T2, which is presently impractical.
>
>
> **(a) hyperparameter defaults**
>
> See general comment.
>
>
> **(b)	analysis of failure conditions**
>
> This is a work in progress.
>
>
> **(c) early-stopping**
>
> Practical strategies include:
> * Stop early if the random baseline is not beaten
> * Start from an upper and lower bound for κ and proceed iteratively, moving from the top or bottom end depending on which direction yields better results.
> * Stop when there is no improvement in the next step.
>
> We did not implement this in the current work because the optimization time is negligible compared with the training time.
>
>
> **(d)	histograms or test sample coverage plots would illustrate the “balanced coverage” claim**
>
> The “balanced” coverage refers not only to distribution across label classes but also to coverage of implicit concepts that are not encoded in the labels (e.g., orange cars when color is not part of the label). If balancing were solely about label frequencies, we could simply retain a representative sample per label, so a histogram of label counts would not capture the intended notion. A more informative diagnostic is the distribution of per-test-sample total value (influence) across test samples. We plan to include such a plot in the revision or supplementary material. However, the practical utility of the approach is reflected in the paper’s plots; a distribution of total per-test-sample value would mainly confirm the intuition behind this.

---

### Official Review · Reviewer_GUf2 · 2025-11-05

**Soundness:** 2
**Presentation:** 4
**Contribution:** 2
**Rating:** 4
**Confidence:** 4

**Summary:**

This paper introduces Constraint-Data-Value-Maximization (CDVM), a novel approach for data pruning that addresses key limitations of existing Shapley-based data valuation methods. The authors demonstrate that semi-value approaches tend to assign lower importance to instances in larger clusters, leading to imbalanced data pruning. To overcome these issues, CDVM formulates data pruning as a constrained optimization problem over a data attribution matrix that tracks training data influence on individual test samples. CDVM achieves state-of-the-art accuracy in 28 out of 36 configurations while maintaining competitive runtime.

**Strengths:**

Very clear presentation.

The motivation for improving the current scoring-based data pruning is very well-motivated.

**Weaknesses:**

It is unclear to me why semivalue-based approaches cannot be used for attribution on individual test samples. While the original Data Shapley and Banzhaf papers employ average test accuracy/loss as their evaluation metric—primarily because they target "data valuation" applications—there is fundamentally little distinction between attribution and valuation. Adapting these methods to compute individual test loss/accuracy appears to require only minor modifications to the code, if I understand correctly. This would enable more direct apple-to-apple comparisons in experiments, such as Data Banzhaf versus Data Banzhaf + CDVM.

The proposed approach is straightforward but lacks theoretical justification. For example, how does the proposed approach compared with the line in coresets (e.g., https://proceedings.mlr.press/v119/mirzasoleiman20a.html)?

The experiment scale is very small. The authors mentioned that "CDVM relies on a selected soft upper bound κ and incurs quadratic cost in computing and storing T (e.g., roughly 250 GB for a naive implementation without sparsity on the full ImageNet-1k train
and val splits)", which is not very clear to me why that's the case. If it means the storage cost during the computation of gradients, here's a highly efficient implementation for computing In-Run Data Shapley (aka TracIn-Ideal) https://github.com/Jiachen-T-Wang/GhostSuite

The following reference is missing, but it seems highly relevant to the discussion in Section 2.1.3.

[1] Wang, Jiachen T., et al. "Rethinking data shapley for data selection tasks: Misleads and merits." ICML 2024

Line 337: citation for Maximum Sample Reuse is wrong.

**Questions:**

See weakness.

---

> ### Author Response · Authors · 2025-11-26
> **Individual concerns of Reviewer GUf2**
>
> Thank you for your time and effort in reviewing. Our response to the concerns raised by multiple reviewers is provided in the general comment above. Regarding individual concerns:
>
> **Why semivalue-based approaches cannot be used for attribution on individual test samples?**
>
> Semivalue-based approaches can be used for attribution on individual test samples. In our setup we construct an attribution matrix T of shape (n_train, n_test), where T[i, j] is the contribution of training instance i to test instance j. This per-test attribution can be computed by applying Data Shapley or Data Banzhaf (or other methods). We use DataBanzhaf due to its performance. DataBanzhaf + CDVM corresponds to applying our optimization algorithm to the attribution matrix T computed with DataBanzhaf, while DataBanzhaf refers to the data value computed by simply averaging over the attribution matrix T.

---

### Author Response · Authors · 2025-11-26
**Response concerning points made by multiple reviewers**

Thank you to the reviewers for their time and feedback. We have split the response into two parts: (i) responses to individual reviewer as comments to the individual review, and (ii) a response addressing points raised by multiple reviewers.

**Scalability, computational cost and large-scale experiments**

We acknowledge concerns about scalability and large-scale experiments. We evaluate our method on training sets of size 1000 largely because this setting is common in the literature (e.g., Banzhaf and OpenDataVal) and is adopted by OpenDataVal.

We also applied the method to a larger dataset, Fashion-MNIST, in Appendix B.3 to illustrate scaling behavior. We did not compare all baselines on this dataset due to time constraints. While Fashion-MNIST is not a large-scale dataset, this experiment demonstrates that CDVM scales to larger data sizes. In Fashion-MNIST, the optimization can be solved in 10–30 minutes thanks to sparsity in the attribution matrix, whereas training enough models to estimate T requires several hours. The latter cost is shared by all methods that estimate semi-value attributions; the overhead introduced by CDVM is primarily the optimization.

It is reasonable to question whether semi-value-based attributions are the right choice for pruning given the overhead. However, this concern is not unique to our work; prior studies have also precomputed memorization and influence estimates on ImageNet (2,000 ResNet-50 models, each trained on a random 70% subset of the full ImageNet training set), though they did not share the full attribution matrix due to its size. See https://pluskid.github.io/influence-memorization/. We use an even smaller sampling probability when estimating T (e.g., 1–5% of the dataset). Thus, our approach would be faster by roughly a factor of 10.

Finally, we estimate T via retraining because this approach is effective and fast on our benchmark datasets. TRAK (arXiv:2303.14186) is a recent method that computes data attributions without retraining as many models. It could be used to estimate attributions for larger datasets, but we did not use it because it would add another layer of uncertainty without delivering runtime benefits for our benchmarks. For large datasets, however, TRAK could be a viable option.

A second bottleneck on large-scale datasets is the size of the attribution matrix T (e.g., ~250 GB for full train × test attributions on ImageNet). This size can be reduced by exploiting sparsity (e.g., on Fashion-MNIST we keep less than 10% of entries, reducing the size to roughly 25 GB). It could be further reduced by using half-precision floats, and the optimization could be solved iteratively on smaller chunks if needed.

We acknowledge that the scalability discussion could be overlooked and that it should be presented more clearly. We will address this more explicitly with the additional page.


**Theoretical grounding and related literature**

We acknowledge that the paper could benefit from a more explicit discussion of related literature. We are aware of the relationship to core-sets, active-learning and other methods, and prior work has benchmarked against some of these (e.g., https://arxiv.org/abs/2206.14486, https://arxiv.org/abs/2205.09329 ). Space constraints in the main text led us to omit a detailed treatment. Our primary comparisons so far have been with memorization and InflOpt, which have demonstrated state-of-the-art performance on pruning benchmarks in prior work (e.g., https://arxiv.org/abs/2206.14486). We agree that clarifying the connections and distinctions between CDVM and core-set approaches is important, and we will add an extended discussion.

**Hyperparameter sensitivity**

We acknowledge that introducing new hyperparameters is a downside of our method. α mainly trades off the data-value term against the size constraint; in our experiments α tends to be robust, with common choices around 0.3 or 0.6, and α = 0.5 is a reasonable default

κ is more dataset- and subset-size specific. As shown in Figure 4c, κ generally increases with larger retained subset sizes, which is plausible since the per-test-influence sum grows with subset size. The structure of the attribution matrix also matters: if many entries are near zero, a smaller κ may be appropriate.

As a rule of thumb, we initially used κ = max(T) + |S| · mean(T) to tailor κ to the dataset and budget. On the small datasets used in our experiments, the optimization completed in under 1 second, so we performed the grid search over κ (and α) to refine the results. In Figure 4(a), CDVM without grid search scores slightly below CDVM with a 5k budget (e.g., 0.90 vs. 0.92 on the normalized metric), while DataOOB is the second-best baseline at around 0.65. Therefore, although grid-search provides small improvements, the main gain comes from the CDVM formulation itself.

---

### Note · Authors · 2026-01-12

**Comment:**

We thank the reviewers for their time and constructive feedback on our work. We have decided to withdraw the submission due to ongoing revisions and a planned resubmission to a different venue.

**Withdrawal Confirmation:**

I have read and agree with the venue's withdrawal policy on behalf of myself and my co-authors.